# Microwave Assisted Green Synthesis of Silver Nanoparticles Using Mulberry Leaves Extract and Silver Nitrate Solution

**Le Ngoc Liem [1],* , Nguyen Phuoc The [2] and Dieu Nguyen [2]**

[1]  Office of the Board of Trustees, Duy Tan University, 03 Quang Trung, Danang 550000, Vietnam
[2]  Faculty of Natural Sciences, Duy Tan University, 03 Quang Trung, Danang 550000, Vietnam;
    nguyenphuocthe@dtu.edu.vn (N.P.T.); nguyentxuandieu@dtu.edu.vn (D.N.)
*   Correspondence: lengocliem@dtu.edu.vn; Tel.: +84-915-995-376

**Abstract:** In this work, silver nanoparticles (AgNPs) were synthesized quickly and in an eco-friendly manner using the extract of Mulberry leaves and aqueous solution of silver nitrate without any toxic chemicals (Yuet et al. *Int. J. Nanomed.* **2012**, *7*, 4263–4267; Krishnakuma and Adavallan. *Adv. Nat. Sci. Nanosci. Nanotechnol.* **2014**, *5*, 025018). The Mulberry leaves extract functions as both a stabilizing and reducing agent. The UV-Vis spectroscopy shows a peak maximum at 430 nm. The transmission electron microscopy (TEM) image illustrated of synthesized AgNPs were nearly spherical-shaped particles whose sizes range from 15 to 20 nm. The TEM image of Nano Silver solution sample synthesized by the microwave assisted method shows nearly spherical particles, with an average particle size estimated at 10 nm. The absorption UV-vis spectrum of silver nanoparticles synthesized by the microwave assisted method (AgNPsmw) shows a sharp absorption band around 415 nm. The UV-Vis spectrum of AgNPsmw after two months of storage shows negligible peak changes of silver nanoparticles.

**Keywords:** silver nanoparticles; mulberry leaves extract

## 1. Introduction

Nanoparticles (NPs) are defined as small particles sized between 1 and 100 nm. Compared with the material bulk states, NPs have gained prominence in recent technological advancements due to their tunable physicochemical characteristics such as their melting point, wettability, electrical and thermal conductivity, catalytic activity, light absorption, better tunable optical properties, and higher reactivity. Various chemical and physical methods have been employed to prepare silver nanoparticles, including chemical reduction electrochemical techniques, and photochemical reduction. Among all the synthetic methods, chemical reduction is most commonly used. However, the chemical synthesis of nanoparticles may lead to the presence of some toxic chemicals. Several studies have focused on green synthesis approaches to avoid using hazardous materials. Synthesizing silver nanoparticles using mulberry leaves extract as a reducing agent not only offers many advantages but also uses less chemicals, thus reducing the pollution caused to the environment. In recent years, nanostructured materials have obtained many applications relevant to daily life. Silver nanoparticles (AgNPs) are used in a wide range of applications, including pharmaceuticals, cosmetics, medical devices, foodware, clothing and water purification, agriculture and in wastewater treatment, etc. due to their antimicrobial properties [1–7]. In this study, mulberry leaves extract had been used as a reduction agent and stabilizing agent. Synthesizing silver nanoparticles using mulberry leaves extract as a reducing agent not only offers many advantages but also uses less chemicals. The green synthesis combined with the microwave assistance make up a highly effective and eco-friendly method.

The mulberry (Figure 1) is a woody plant that grows quickly and within a short proliferation period. There are about 10 to 16 species of genus Morus that are found in the subtropical climates and the warm and temperate regions of Africa and North America [8]. Those species have been cultivated in many Asian countries such as China, India, Korea, Japan, Thailand and Vietnam, where the leaves have been used as food for silkworms [9]. Mulberry is known to be used in some traditional Chinese medicinal formulas. There are several studies have shown that it may provide health benefits. There are many biologically active compounds and many phytochemicals in mulberry leaves. There are also many important pharmacological properties such as antibacterial, antiviral [5–9], antitussive, hypoglycemic, antiatherogenic [10], hypotensive [11], diuretic, astringent, antioxidant [12,13] and $\alpha$-amylase inhibitory effects [10].

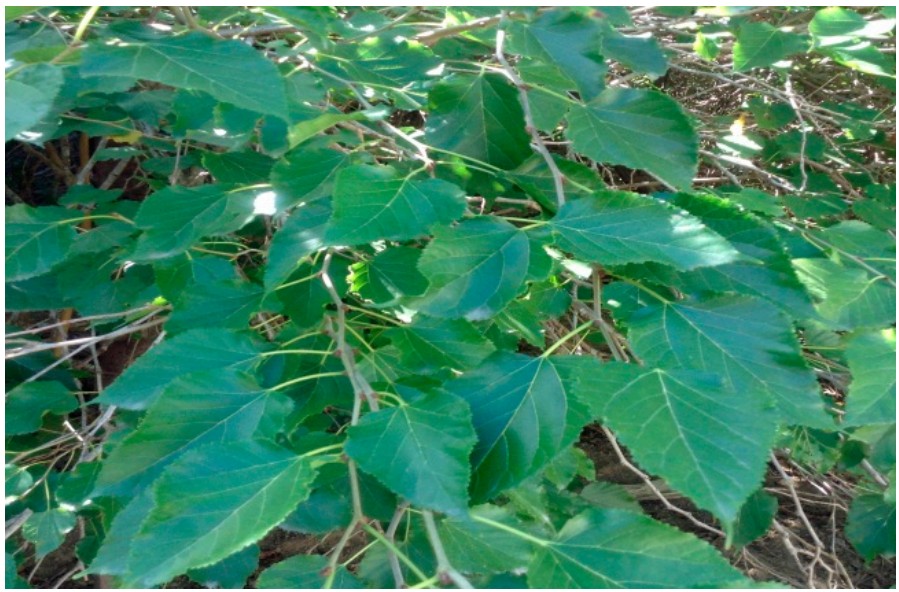

**Figure 1.** Picture of Mulberry trees leaves.

## 2. Materials and Methods

### 2.1. Preparation of Mulberry Leaves Extract and Silver Nitrate Solution

Mulberry leaves were collected from a residential garden house in Hoi An, Quang Nam province, Vietnam. The leaves collected must be intact and at their prime (neither be too young or too old). Those fresh leaves were then cleaned with fresh water and let air dried by laying them out evenly. 10 g of fresh leaves was obtained, cut into thin strips, then placed into a 200 mL heat-resistant glass flask. Then, they were boiled with distilled water in 5 min, cooled and the mixture was filtered with Whatman filter paper using a vacuum filter [4–7]. The mulberry leaves extract had a light yellow color. The extracted solution was stored in a fridge for further use. Dissolved silver nitrate ($AgNO_3$) from Sigma Aldrich was mixed with distilled water to get $4.10^{-3}$ M aqueous $AgNO_3$ solution.

### 2.2. Synthesis of Nano Silver Material

#### 2.2.1. Non-Microwave Assisted Synthesis of Nano Silver

Visual Observation and UV-vis Spectral

$AgNO_3$ $4.10^{-3}$ M solution and mulberry leaves extract were mixed at different ratios as shown in Table 1. These mixtures were placed on a shaker and stirred for 30 min, 150 rpm at room temperature. After 30 min, the color of all the mixtures changed from light brown to dark red, except for mixtures $M_0$ and $M_1$. The color of each mixture varies depending on the ratio of $AgNO_3$ concentration and

mulberry leaves extract that were used (Figure 2). This observation implied that Nano Silver particles were formed.

**Table 1.** Synthesized samples.

| AgNO$_3$ Solution (mL) | Mulberry Leaves Extract (mL) | Samples |
|:---:|:---:|:---:|
| 0 | 10 | M$_0$ |
| 40 | 0 | M$_1$ |
| 40 | 1 | M$_2$ |
| 40 | 3 | M$_3$ |
| 40 | 5 | M$_4$ |
| 40 | 6 | M$_5$ |
| 40 | 7 | M$_6$ |
| 40 | 8 | M$_7$ |

*Characterization Techniques*: In order to investigate the optical properties, we measured the UV-vis absorption spectra using GE Ultrospec 7000 UV-vis spectrophotometer (GE Lifesciences, Freiburg, Germany). Transmission Electron Microscope (TEM) analysis of silver nanoparticles was done using JEOL JEM 1010 (JEOL, Tokyo, Japan). X-ray diffraction spectra were measured by the diffractometer Bruker D8-Advance (Bruker, Karlsruhe, Germany), Fourier transform infrared (FTIR) Spectra. For mulberry leaves, extract was obtained in the range 400–4000 cm$^{-1}$ with IRAffinity-1S Shimadzu FTIR spectrophotometer (Shimadzu, Tokyo, Japan).

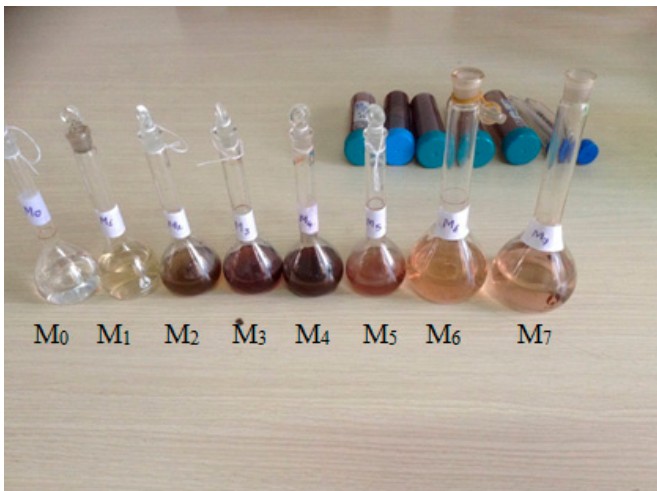

**Figure 2.** Photograph color change of colloids.

The intensity of absorption within the wavelength of 200 to 250 nm is very strong. Therefore, before measuring UV-vis, samples were diluted 40 times. The UV-vis spectrum of the material shows a strong surface plasmonic resonance band centered at 430 nm. For the samples M$_2$, M$_3$ and M$_4$, the absorbance intensity increasesd as we increased the volume of mulberry leaf extract from 1 mL to 5 mL. For samples whose volume of mulberry leaves extract were more than 5 mL, the intensity decreased as the volume of mulberry leaves extract increased. Combining spectral information with qualitative observation, we could conclude that AgNPs has been formed when mixing AgNO$_3$ solution with mulberry leaves extract. Among our samples, the sample M$_4$ not only showed the highest peak intensity value but also had the darkest red color; the observations indicated that it had the most AgNPs particles. We can also infer that, at high concentration of mulberry leaves extract, the rate of AgNPs production is so rapid that it prevents the formation of protective layer between particles. For this to happen, the aggregation phenomenon between particles occurred and increased the particle size. The increased particle size reduced peak intensity and shifted the peak to a longer wavelength (as

in $M_5$, $M_6$ and $M_7$ samples). The absorption spectrum of $AgNO_3$ aqueous solution, shown in the olive colored line in Figure 3, had no peak in the measuring range. The absorption spectrum of mulberry leaves extract, shown in orange line in Figure 3, had absorption peak in the short wavelength region due to an organic substance found in the extract solution and had no peak at wavelengths longer than 400 nm. The spectral information indicated that AgNPs particles were formed only when $AgNO_3$ was mixed with mulberry extract. When magnified from 380 to 700 nm of the absorption spectrum (Figure 3), discrete lines were shown.

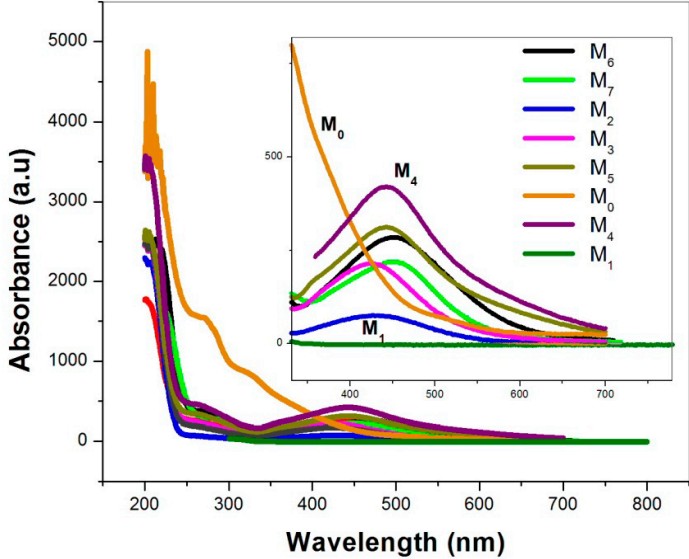

**Figure 3.** UV-Vis spectra of $AgNO_3$ solution (M1), Mulberry extract (M0) and AgNps prepared at different Mulberry leaves extract.

2.2.2. Microwave Asissted Systhesis of Nano Silver

Visual Observation and UV-vis Spectral

We mixed 50 mL of $AgNO_3$ solution with 6 mL of mulberry leaves extract and divided it into 2 equal portions. Then, we put the first portion ($M_8$) on a shaker machine and stirred for 30 min, 150 rpm at room temperature. The other portion ($M_9$) was heated in a microwave for one minute. The changed colors of $M_9$ from light yellow to dark red within one minute in the microwave indicated that the efficiency of (AgNPsmw) synthesis using the microwave assisted method was higher than the non-microwave assisted method. The sample colors are shown in Figure 4.

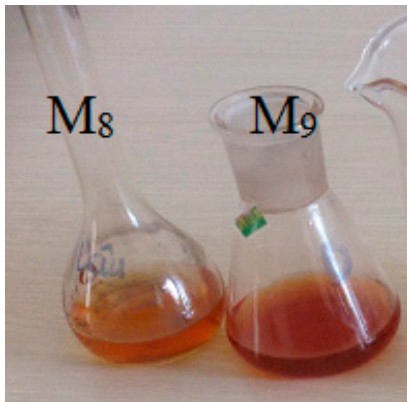

**Figure 4.** Photograph of silver nanoparticles (AgNPs) ($M_8$) and of AgNPsmw ($M_9$).

The UV-vis spectra (Figure 5) shows that absorption intensity at around 415 nm of $M_9$ is higher than that of $M_8$ and the maximum wavelength of $M_9$ is smaller than the $M_8$. After two months of storage, the $M_9$ sample was renamed $M_{10}$. Compared to the spectrum of $M_9$, the UV-vis spectrum taken for $M_{10}$ (the blue line of Figure 5) shows a slight decrease in the peak intensity and the peak shift of 5 nm towards a longer wavelength. The above observations indicated that the synthesized colloid solution is highly stable. Compared with the work of other authors, our study resulted in a smaller particle size, short-time synthesis, and a higher absorption peak intensity.

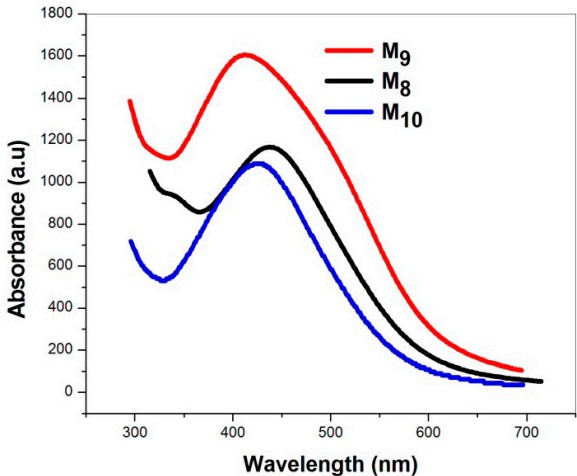

**Figure 5.** UV-Vis spectra of AgNPs ($M_8$), of AgNPsmw ($M_9$) and of AgNPsmw after two months of storage ($M_{10}$).

*2.3. X-ray Diffraction (XRD) Studies*

Cloth was cut into 25 $cm^2$ pieces and dipped into a colloids solution of AgNPs for 10 min, and another piece of cloth was not dipped into a colloids solution of AgNPs. The XRD pattern of the cloth dipped into a colloids solution AgNPs and the cloth not dipped into a colloids solution AgNPs are shown in Figure 6. The XRD pattern of the undipped cloth showed no characteristics of peak silver. On the other hand, the XRD pattern of the cloth that was dipped in the colloids AgNPs showed the characteristic of peak silver. Four main characteristics of the diffraction peak for Ag were observed at 2θ values of 38.2°, 44.1°, 64.5°, and 77.6°, which correspond to the (111), (200), (220) and (311) crystallographic planes of face-centered cubic (fcc) Ag crystals, respectively (JCPDS 00-004-0783) [14,15]. Because the AgNps concentration was low and the cloth's internal structure had lots of empty spaces, not many AgNPs particles were deposited onto the cloth surface, thus the peaks intensity was low.

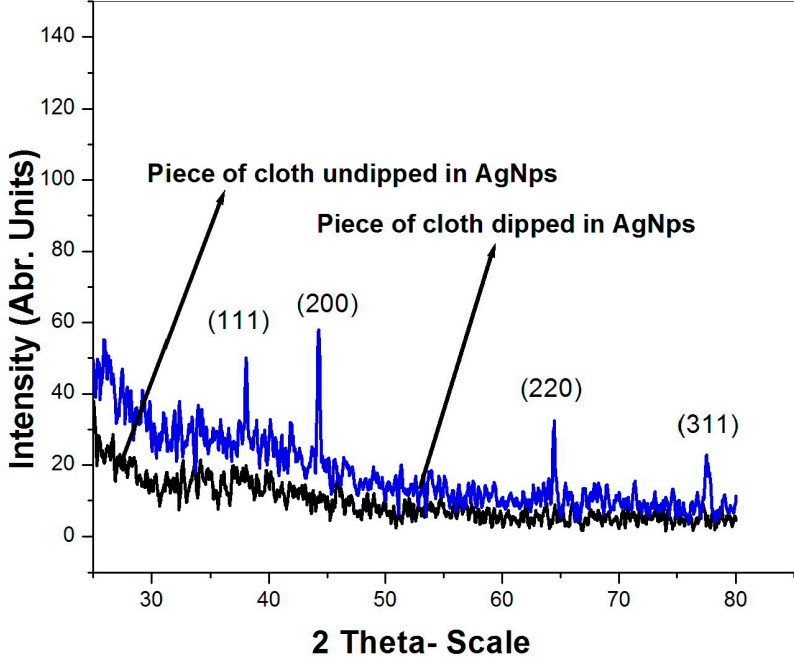

**Figure 6.** X-ray diffraction of a pieces of cloth dipped into a colloids solution AgNPs (blue line), and a pieces of cloth not dipped into a colloids solution AgNPs (black line).

### 2.4. Transmission Electron Microscope (TEM) Analysis

The size and shape of the AgNPs was further confirmed by TEM analysis, is shown in Figure 7. The TEM image of $M_8$ showed a relatively uniform spherical particle size, ranging from 15–20 nm. The TEM image of $M_9$ showed particle size about 10 nm. The particle sizes were more uniform and no sign of nanoparticle clustering was observed.

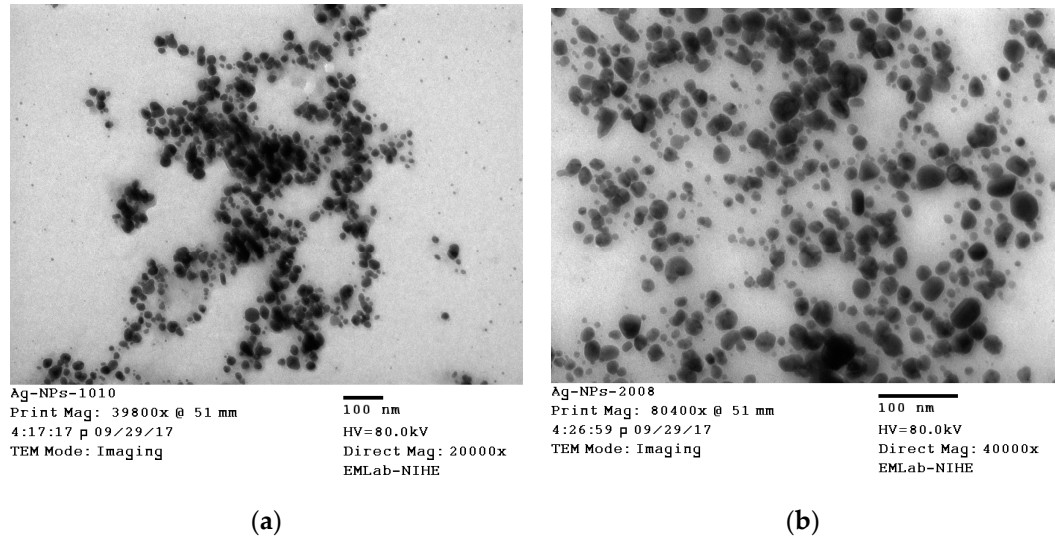

(**a**)                               (**b**)

**Figure 7.** Transmission Electron Microscope (TEM) images of silver nanoparticles ($M_8$ and $M_9$).

Figure 8 shows the histogram of size distribution of silver nano particles. The average particle size measured from the TEM image is 10 nm. This large variation in particle size was due to the presence of a few irregular shaped particles.

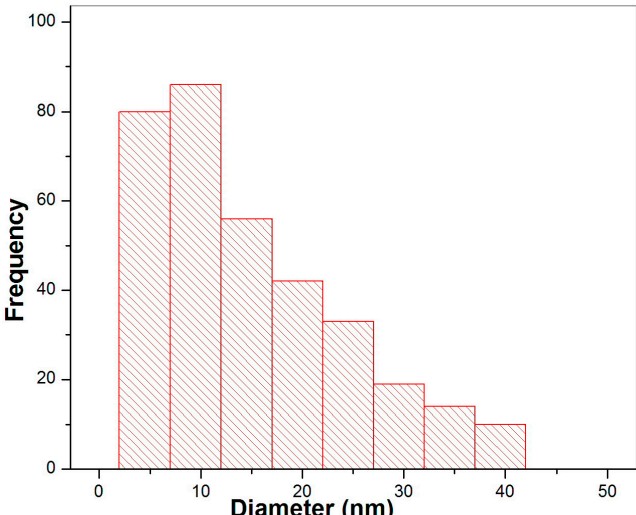

**Figure 8.** Histogram showing the particle sizes of AgNPs corresponding to TEM images $M_9$.

## 2.5. FT-IR Spectrum

0.5 mL of the $AgNO_3$ solution was dropped into 50 mL of mulberry leaves extract to prevent the extracted solution from rotting before taking the spectrum for mulberry leaf extract (From the sample synthesis to the FT-IR spectrum takes about 5 days). An FT-IR spectrum of silver nanoparticles synthesized by this green method is shown in Figure 9. A number of absorption peaks at 3261 cm$^{-1}$ and 1637 cm$^{-1}$. The peaks at 3261 cm$^{-1}$ corresponds to O-H and N-H bonds, the peak at 1637 cm$^{-1}$ corresponds to the C=O bond, indicating the biomaterial bind to the silver nanoparticles through amine and C=O of amide I and amid II of the protein [1,6,7,16]. These results indicate that mulberry leaves extract acts as a reducing and stabilizing agent for silver particles. According to studies by A.K. Mittal et al., the extract of Dhatura metel contained alkaloids, proteins, enzymes, amino acids, alcoholic compounds, and polysaccharides, which were said to be responsible for the reduction of the silver ions to nanoparticles. Quinol and chlorophyll pigments present in the extract also contributed to the reduction and stabilization of the nanoparticles [17]. Leaf extracts of Mulberry also contain similar leaf extracts of Dhatura metel [18]. Therefore, mulberry leaf extract also played a role in the reduction and stabilization of the nanoparticles.

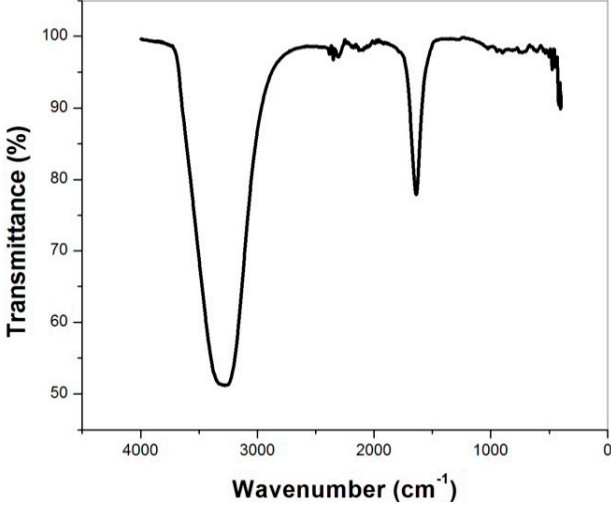

**Figure 9.** FTIR spectrum of mulberry leaves extract.

## 3. Conclusions

The silver nanoparticles (AgNPs) colloid solution has been successfully synthesized, using a green and eco-friendly method. This is a quick, highly effective and less chemical-consuming method. The Microwave assisted green synthesis method is more effective than the non-microwave assisted method. The UV-Vis spectrum of AgNPs has an absorbance peak ranging from 425 nm–435 nm. Nano-silver particles are spherical shaped with size ranging from 15 nm to 20 nm. UV-vis spectrum of AgNPsmw shows peak at 415 nm, with an average particle size of 10 nm. The results observed after two months of storage of AgNPsmw are quite stable.

**Author Contributions:** Conceptualization, L.N.L.; Methodology, L.N.L. and N.P.T.; Validation, L.N.L. and N.P.T.; Formal analysis, L.N.L.; Investigation, L.N.L., D.N. and N.P.T.; Writing—Original draft preparation, L.N.L.; Writing—Review and Editing, L.N.L., D.N. and N.P.T; supervision, L.N.L.

**Funding:** This research received no external funding.

**Conflicts of Interest:** The authors declare no conflict of interest.

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
