# Peer review of "Microwave Assisted Green Synthesis of Silver Nanoparticles Using Mulberry Leaves Extract and Silver Nitrate Solution"

_technologies, doi:10.3390/technologies7010007_

Reviewer 1 Report

1. The abbreviations of silver nanoparticles should be uniformed: “AgNPs” or “AgNps”.

2. The titles of Figs. 3 – 6 are not clear and should be supplemented.

3. Generally, in conclusions research results are summarized. However, the statement in conclusions “Synthesizing silver nanoparticles by mulberry leaves extract is eco-friendly and good antimicrobial efficiency against bacteria, viruses and other microorganisms [16] and used in wastewater treatment, due to their antimicrobial properties” is not supported by any research results.

Author Response

Thank you for your review of our paper. We have answered each of your points below:

Point 1: The abbreviations of silver nanoparticles should be uniformed: “AgNPs” or “AgNps”.

Response 1: The abbreviations of silver nanoparticles were be uniformed: "AgNPs"

Point 2: The titles of Figs. 3 – 6 are not clear and should be supplemented.

Response 2: The titles of Figs. 3 – 6 were be supplemented

Point 3:  Generally, in conclusions research results are summarized. However, the statement in conclusions “Synthesizing silver nanoparticles by mulberry leaves extract is eco-friendly and good antimicrobial efficiency against bacteria, viruses and other microorganisms [16] and used in wastewater treatment, due to their antimicrobial properties” is not supported by any research results.

Response 3: We have deleted “Synthesizing silver nanoparticles by mulberry leaves extract is eco-friendly and good antimicrobial efficiency against bacteria, viruses and other microorganisms [16] and used in wastewater treatment, due to their antimicrobial properties”

I look forward to your comments.

Best regards,

Reviewer 2 Report

The authors have addressed most of the reviewers comments, now the manuscript is acceptable for the publication. But, still the manuscript need English corrections.

Author Response

Thank you for your review of our paper. We have answered each of your points below

Point 1:The authors have addressed most of the reviewers comments, now the manuscript is acceptable for the publication. But, still the manuscript need English corrections”

Response 1: We have edited of English language and style required. I look forward to your comments.

We hope that these revisions improve the paper such that you and the reviewers now deem it worthy of publication in  Technologies.

Best regards,

This manuscript is a resubmission of an earlier submission. The following is a list of the peer review reports and author responses from that submission.

Round  1

Reviewer 1 Report

The authors report a method for synthesis of silver nanoparticles. The science presented is sound. There are several grammatical mistakes that need to be fixed. The authors should edit the papers to fix those mistakes.

Reviewer 2 Report

In this work silver nanoparticles obtained using microwave-assisted green synthesis. The subject of the research is not new and widely investigated. Some comments:

1. Fig. 1 is uninformative and, therefore, dispensable.

2. Obtained Mulberry leaves extract should be characterized; i.e. phytochemical analysis can be carried out.

3. P. 3, line 66: How to understand proposition ”… each samples were dissolved 40 times“. It should be explained.

4. P. 3, line 75: “… the formation of protective layer between particles”: What protective layer is formed around particles? What is its composition?

5. Silver nanoparticles size distribution histogram should be presented.

6. In this paper an antimicrobial efficiency wasn’t investigated, therefore, it should not be stated that synthesized silver nanoparticles could be used as an antimicrobial agent or in wastewater treatment.

Reviewer 3 Report

The manuscript “Microwave Assisted Green Synthesis of Silver Nanoparticles Using Mulberry Leaves Extract and Silver Nitrate Solution” submitted to Technologies demonstrates the green synthesis of silver nanoparticles using Mulbery leaf extract. The objectives of the research are well-founded, as currently nanotechnology is a fast-developing industry, posing substantial impacts on economy, society and environment. The vast production, use, and disposal of nanomaterials may have long lasting effect on the environment. In this context, the green synthesis of nanomaterials is highly desirable. In this study, silver nanoparticles have been prepared using the leaf extract. However, this work lacks sufficient novelty and similar kind of work has already been published (Awwad, Akl M., and Nidà M. Salem. "Green Synthesis of Silver Nanoparticles byMulberry LeavesExtract." Nanoscience and Nanotechnology 2.4 (2012): 125-128). The characterizations of as-prepared samples were carried out through various techniques which are sufficient but the results presented and discussed are not good enough to be accepted. Particularly, the XRD, data is very poor.  Furthermore, in the submitted manuscript there are some other drawbacks which, in the opinion of the reviewer, seriously impact the quality of the work. For instance, the language used in this manuscript and the grammar has to be further improved; the introduction part must be revised to include more information regarding the importance of silver NPs and the green synthesis methods. Therefore, the work lacks merits for the publication and should be rejected. Besides, the reviewer has also noticed some other mistakes which are as following.

Technologies EISSN 2227-7080 Published by MDPI AG, Basel, Switzerland RSS E-Mail Table of Contents Alert
Back to Top